# Future Is Unlicensed: Private 5G Unlicensed Network for Connecting Industries of Future

**DOI:** 10.3390/s20102774

**Published:** 2020-05-13

**Authors:** Rojeena Bajracharya, Rakesh Shrestha, Haejoon Jung

**Affiliations:** 1Department of Information & Telecommunication Engineering, Incheon National University, Incheon 22012, Korea; rojeena@inu.ac.kr; 2Yonsei Institute of Convergence Technology, Yonsei University, Incheon 21983, Korea; rakez_shre@yonsei.ac.kr

**Keywords:** Industry 4.0, unlicensed band, NR-U, shared spectrum, 5G

## Abstract

This paper aims to unlock the unlicensed band potential in realizing the Industry 4.0 communication goals of the Fifth-Generation (5G) and beyond. New Radio in the Unlicensed band (NR-U) is a new NR Release 16 mode of operation that has the capability to offer the necessary technology for cellular operators to integrate the unlicensed spectrum into 5G networks. NR-U enables both uplink and downlink operation in unlicensed bands, supporting 5G advanced features of ultra-high-speed, high bandwidth, low latency, and improvement in the reliability of wireless communications, which is essential to address massive-scale and highly-diverse future industrial networks. This paper highlights NR-U as a next-generation communication technology for smart industrial network communication and discusses the technology trends adopted by 5G in support of the Industry 4.0 revolution. However, due to operation in the shared/unlicensed spectrum, NR-U possesses several regulatory and coexistence challenges, limiting its application for operationally intensive environments such as manufacturing, supply chain, transportation systems, and energy. Thus, we discuss the significant challenges and potential solution approaches such as shared maximum channel occupancy time (MCOT), handover skipping, the self-organized network (SON), the adaptive back-off mechanism, and the multi-domain coexistence approach to overcome the unlicensed/shared band challenges and boost the realization of NR-U technology in mission-critical industrial applications. Further, we highlight the role of machine learning in providing the necessary intelligence and adaptation mechanisms for the realization of industrial 5G communication goals.

## 1. Introduction

The term “wireless networking” means widespread connectivity, but we have yet to see that in today’s industrial and business-critical domains such as manufacturing, supply chain, transportation systems, and energy [1,2]. The existing mobile networks designed and deployed for enterprise applications provide opportunities to optimize and redefine business within the limitations of wired (Ethernet/optical) and WiFi networks. For industrial applications, the ability to design mobile networks to meet the coverage, performance, and security requirements of critical applications (such as mobile robots, automated guided vehicles, and head-mounted displays with advanced mobile applications for workers) is fundamental to the new wave of cyber-physical systems known as Industry 4.0 [3,4]. In its support, a private LTE network that is utilizing dedicated radio equipment to serve the premises of local and independent networks (such as the industrial network) with specific Internet of Things (IoT) applications and services is undergoing investigation under 5G technology [5,6]. A private network is the Long-Term Evolution (LTE)-based wireless communication technology for local/independent networks, which resolves user concerns of reliability and service quality, as well as security and compliance through a single, scalable wireless networking deployment leveraging LTE’s technology. However, a private LTE network operating in a licensed spectrum requires a license from the national regulator or an agreement with a license holder, typically a mobile operator [7]. Thus, the use of a scarce licensed spectrum model is limiting the flexibility of private LTE to operate anywhere and everywhere, without a costly spectrum or specialists with expertise in network deployments [8]. Driven by the limited availability of the licensed spectrum, MulteFire [9] and NR-U technology were introduced by 4G in Release 14 [10] and 5G in Release 16 [6], respectively, to generate an opportunity to alleviate the spectrum crunch in future wireless networks by operating in the unlicensed band. MulteFire makes the deployment of standalone LTE [11] radio technology possible in the 5 GHz unlicensed band, whereas 3GPP proposes 5G New Radio in the Unlicensed band (NR-U), an idea of extending MulteFire in 5G with both standalone and non-standalone modes (anchor with licensed spectrum) [12] of operation in the shared/unlicensed bands. We believe both technologies are potential candidates for next-generation communication for a smart industry that merges the quality of performance of LTE with the deployment ease of WiFi [13]. That is, end-users (enterprises) will install MulteFire/NR-U access points in place of WiFi gateways to provide LTE coverage for the Industry 4.0 scenario. In Table 1, we list the use cases of NR-U in the industrial domain.

In this paper, we argue that the unlicensed spectrum plays a crucial role in realizing the industrial communication goals of 5G and beyond by offering ultra-high-speed, high capacity, low latency, and improvement in the reliability of wireless communications, which is essential to address massive-scale and highly-diverse future enterprise networks [15]. However, the unlicensed spectrum places the challenge of “coexisting networks” in a shared band [16], which mostly lack the means of coordination and observation. The operation in the unlicensed spectrum is subject to various limitations and restrictions [6], which are regional and band-specific, highly reducing the efficiency of LTE technology. The typical limits are in terms of transmit power, channel bandwidth, power spectral density, and the transmit duration that can be used by each device. In addition, sharing protocols may also be specified in some bands to protect other systems in the band or to allow efficient sharing [17]. Dynamic frequency selection was proposed to safeguard radar systems in the 5 GHz band, and listen-before-talk (LBT) [18] was introduced to enable efficient spectrum sharing by minimizing inter-user interference in the unlicensed spectrum. Both proposed protocols highly degrade the LTE performance in terms of latency and throughput. This issue is further aggravated by the heterogeneity, massive connectivity, and ubiquity of industrial network systems and applications [19]. Therefore, 5G NR-U needs to provide mechanisms to coexist and even converge through necessary modifications to the LTE specification to combat the loss from the regulatory requirements. In this paper, we open up 5G NR-U prospects specifically for mission-critical industrial applications. Moreover, for the first time, we point out the possible challenges that arise from the use of New Radio in the Unlicensed/shared band in industrial applications and suggest possible research directions to solve the challenges. We envision machine learning as a toolbox, which will play a fundamental role in making networks adapt to the coexistence setting by maximizing unlicensed spectrum efficiency, easing the capacity constraint for enterprises’ 5G network and beyond.

The rest of this paper is organized as follows. The brief overview of Industry 4.0 and NR-U technology is described in Section 2. The prospects of 5G NR-U in industrial applications are discussed in Section 3. The main technical challenges and research directions for industrial NR-U are presented in Section 4. Finally, conclusions are drawn in Section 5.

## 2. Background

### 2.1. Industry 4.0

The advent of modern emerging industrial technologies, known as Industry 4.0, is an evolutionary trend of automation and data exchange in industrial technology after mechanization (First Industry), electrification (Second Industry), and digitalization (Third Industry) [20,21]. The fourth industrial revolution focuses on incorporating IoT (Industrial IoT or IIoT [22]) and cyber-physical technologies in the domain of industrial automation with the key purpose of increasing the competitiveness of industrial automation verticals such as manufacturing, energy, utilities, oil and gas refineries, and transportation [23]. For example, in manufacturing systems, we need reliable communication for smart production, maintaining sustainable supply chain networks, and real-time process monitoring, as presented in [4,24]. Communication for automated and connected industries needs strict criteria in terms of latency, availability, throughput, and the number of devices for diverse Industry 4.0 equipment, as shown in Table 2 [25]. Such vital communications needs in Industry 4.0 generate the opportunity for the communication to handle mobile robots, autonomous guided vehicles, and specialized head-mounted displays’ application for employees. In support, 5G NR access technology is actively working on ultra-reliable low-latency communications (URLLC), enhanced mobile broadband (eMBB) for use cases like extended reality (XR), connectivity for massive machine-type communications (mMTC), time sensitive networks (TSN) and Ethernet replacement, and expansion to the unlicensed spectrum for the Industry 4.0 applications [26].

The 3rd Generation Partnership Project (3GPP) introduced NR-U in the unlicensed sub-7 GHz bands in 3GPP Release 16 [6] to provide a necessary technology supporting the future industry and society. NR-U is a groundbreaking transformation of LTE in the Unlicensed band (LTE-U) [28]/Licensed Assisted Access (LAA) [29] from 4G LTE to 5G NR and provides an alternative to mitigate the bandwidth scarcity problem by leveraging the unlicensed spectrum in NR operations. Due to its high availability and relatively easy access to the unlicensed spectrum, NR-U is expected to play a crucial role in future cellular networks. Figure 1 shows the evolution of NR-U from early 2013 as LTE-U, which eventually will lead to the integral part of NR in the future.

### 2.2. NR-U

The 3GPP has defined three main industrial deployment scenarios for NR-U: Carrier Aggregation mode, Dual Connectivity mode and Standalone mode. Carrier Aggregation mode is based on LTE-LAA in Release-13 [30]. In LTE-LAA, the unlicensed spectrum is used to augment downlink capacity, whereas the uplink and control plane signals are passed via the license carrier. Dual Connectivity mode is based on the design of LTE-extended LAA (eLAA) in Release-14 [31], which supports both uplink and downlink data-plane traffic over the unlicensed spectrum, whereas the control plane is anchored over the licensed band. Standalone mode is a novel approach in Release-16 [32,33]. In the first two modes, NR-U is anchored to both licensed band. In standalone NR-U, similar to the MulteFire Alliance for standalone operation [11,34], NR-U is expected to work in the unlicensed spectrum without being anchored to any licensed carrier. Thus, NR-U is attracting much interest in industrial networks since it allows removal of the dependency on public networks. In Table 3, we present a taxonomy of the different deployment modes of NR-U, including the standardization body, operational frequencies, the frequency band, the protocol, the maximum supported bandwidth, supported streams, and the underline technology.

There are also two main modes of spectrum sharing proposed for NR-U in unlicensed scenarios [33,35] as shown in Table 4. The first is asynchronous shared spectrum mode for NR-U. This is the evolutionary path that works within the existing coexistence rules applied to the unlicensed spectrum, i.e., when deployed, NR-U must follow LBT channel access protocols. Thus, NR-U uses a variety of strategies to boost its reliability and availability performance in these shared spectrum bands. Second is the synchronous shared spectrum for NR-U. The synchronous shared spectrum is a more advanced approach of spectrum sharing technology, which allows overlapping of the contention window (CW) of the node. It offers great potential for more efficient spectrum sharing, with more stable, more consistent performance. However, this approach requires sharing mechanisms with a common synchronization reference among operators and will need support from regulators. This synchronous method was popular in the U.S. initially, and now, Europe is also interested in this approach due to the following advantages over asynchronous mode:Improves service predictability and user experience via CW overlapping to minimize the latency.Enables advanced technologies such as coordinated multi-point (CoMP) to enhance spectrum efficiency and reliability.Facilitates flexible sharing via a reservation slot structure, which ensures that no node is blocked from accessing the medium for an extended period, and channel use is “guaranteed.”Enables advanced sharing such as spatial domain multiplexing to improve spectrum utilization.

The NR-U study item has considered different unlicensed bands or shared bands such as 2.4 GHz, 5 GHz, 6 GHz, and 60 GHz [12,36] as initial candidates; however, later on, 3.5 GHz and 37 GHz bands were open for shared access in the U.S. The 3GPP classifies these bands for NR-U as sub-7 GHz and mmWave bands. Sub-7 GHz bands include the 2.4, 3.5, 5, and 6 GHz bands; meanwhile, mmWave bands encompass the 37 and 60 GHz bands [10]. As confirmed by 3GPP, the 5 and 60 GHz bands are attractive candidates for NR-U, since they are currently not very crowded as 2.4 GHz and can offer a large amount of contiguous bandwidth. They both have common global availability and for which most major geographical areas worldwide have authorized the use of wide unlicensed spectrum bandwidth. Table 5 describes the frequency characteristics.

## 3. Opportunities of NR-U for Industry 4.0

WiFi has always been famous for industrial applications due to its easy-to-use and fast deployment characteristics. In many cases, WiFi remains an excellent solution to some extent. However, the use of private LTE over WiFi networks for industries gives the advantage of radio performance in terms of range/link budget, spectral efficiency/capacity, configurable QoS, mobility, interoperability, high to low rate scaling, spectrum options, security, and road map to 5G and beyond. Looking at the evolving practice of LTE, the technological revolution of 4G seemed to be more focused on richer multimedia content for the individual users, while the NR-U technology of 5G holds the potential to revolutionize the industry. Industries and enterprises have always been in the priority group for adopting the leading technology. Hence, 5G NR-U tends to transform the future cyber-physical industries in the following ways (Figure 2).

### 3.1. Reliability

Cellular technology is the most reliable communication technology in wireless technology including WiFi. Hence, the ability to operate a private network in the newly available broadband unlicensed spectrum (3.5/5/6/60 GHz) under cellular technology will offer vastly more reliable means of communication for the industry. Additionally, connectivity between the access points is always coordinated in the cellular system, meaning that there will be a seamless connection as a system transfers from one access point to another. Mission-critical technologies that require continuous, reliable connections, such as safety communications and warning systems, will be supported with extremely trustworthy links.

### 3.2. Latency

One of the critical problems with smart industry application is latency/delay in the time it takes to reach the destination. Latency can make a huge difference when it comes to devices involving near-real-time communication (e.g., physical or virtual reality). The latency that one would predict from 4G networks today is around 50 ms–70 ms (0.05 s). It is reduced to 2 ms–5 ms with 5G and a potential target latency of 1ms with enhancements made in the radio access network (RAN). Moreover, with too many applications and business data stored in the cloud, data rates and delay have become more significant than ever. The NR-U commercial networks open up new doors for such delay-sensitive applications such as robots’ sensors, the systems requiring a high data rate like AI/AR, and those integrating both areas such as telemedicine and public health.

### 3.3. Device Density and QoS

The introduction of IoT devices for automation purposes in factories, offices, and warehouses has increased the number of sensors per square meter, making a high capacity 5G network almost mandatory for industrial applications. 5G networks have promised to accommodate one million devices per square kilometer. Moreover, the present wireless communicating systems and computers available in the industrial field still lack the level of QoS required for mission-critical applications. Thus, with NR-U, system performance and resource use for different mission-critical services can be tailored to the specific needs within the private network deployment (network slicing). For example, the traffic coming from a UAV will be on a different (and higher priority) network slice than the one serving visitors in an office.

### 3.4. Ease of Deployment

For the speedy adoption of new technology, the technology must possess devices that are cost-effective, fast to deploy, and simple to operate. WiFi is one of the best examples of this operation. With NR-U, 5G is now close to replicating this. 5G NR-U no longer requires a costly and scarce license band to transmit; instead, it uses an unlicensed band, which is a free band. Additionally, for enterprise deployment, it does not necessitate any licensing process, for which it can take months to acquire permission. Thus, NR-U service deployment time can be significantly shortened to a matter of days or hours. 5G also introduces the core-network-in-a-box virtualized functions, which offer remote configuration and management. These features not only help in the implementation speed of new services and new business models, but also in the dynamic scalability of network capacity. The use of common equipment such as access points, Ethernet cables, etc., for NR-U deployment has made NR-U deployment as simple as WiFi.

### 3.5. Security

The 5G NR-U network is expected to have full end-to-end protections and must ensure that the information and resources of the industry are protected from attacks and threats. This includes three core concepts of security: confidentiality, integrity, and availability. The use of unlicensed private networks provides NR-U with the necessary network isolation, data protection, and device/user authentication to secure key properties. Moreover, NR-U offers classical subscriber identification module (SIM)-based security, as well as non-SIM-based security for its devices. There is significant deployment flexibility, allowing for local credential management or centralized/remote credential management using a roaming-based solution.

## 4. Challenges and Research Directions

LAA has established a solid foundation for bringing LTE to unlicensed bands. Standalone NR-U will open the door to further innovation and novel use cases as we tackle the associated design challenges. A standalone system must satisfy the following primary requirements to be viable for operation in the unlicensed band: First, it must adhere to any applicable regulatory requirements for unlicensed band operation [39,40], such as limits on transmit power, spectral density, and channel occupancy. Secondly, it must achieve fair coexistence with other nodes of either the same or a different technology, e.g., WiFi. Thus, the realization of standalone NR-U possesses various regulatory and coexistence challenges from the industrial, as well as the technological perspective. Therefore, we categorized the standalone NR-U’s research challenge into following sub-subsections with possible research directions.

### 4.1. Challenge 1: LBT Is Mandatory to Operate in the Unlicensed or Shared Band

“Listen before talk” has been designed to ensure a better neighbor to WiFi while coexisting in the unlicensed band [41]. However, the presence of LBT in channel access, characterized by a decentralized and asynchronous random access to the radio channel for uplink data transmission, can highly degrade NR-U performance in terms of channel access delay and latency due to the LBT requirement for each new transmission. In this scenario, the UEs’ system has a disadvantage in accessing the channel as they need to go through the following contentions for uplink in the NR-U system [13,34]:To access UL service for data transmission, the UE must first send a scheduling request (SR) to the serving eNodeB (eNB). To do this, the UE must first conduct LBT to obtain the medium before transmitting the SR.Once the SR is received, the eNB prepares a UL grant for a specific sub-frame(s) to the UE, and the eNB will again obtain the channel by LBT.After receiving a UL grant, the UE again performs LBT to acquire the channel for UL data transmission.

Furthermore, in the case of a UE having received a UL grant and LBT being needed but failing (time constraint), i.e., the UE loses its opportunity to transmit, the associated frequency/time domain resources for the SR and UL grant transmission are wasted. The UE can perform a transmission for the same data only after the eNB detects that the expected transmission has failed and re-schedules the same data transmission, resulting in an increase in overhead and a delay in transmitting data packets through the uplink. Hence, the uplink LBT procedure needs to be redesigned to improve the UL performance in standalone NR-U, and it must deliver downlink (DL) and uplink (UL) throughputs comparable to or better than other unlicensed technologies.

We already know that the use of LBT in the uplink transmission significantly increases channel access delay due to the grant and scheduling request (SR) required in LTE mode. The main issue observed here is that the UE suffers from long delays before it can attempt to access the channel, which is caused by three LBTs needing to succeed before the UE can transmit. Most ongoing works have proposed to eliminate the granting of access in uplink transmission as in [34,42,43]. However, another simple approach to reduce the overhead of the UL grant transmission mode would be to reduce the grant and LBT transmissions as much as possible as presented in [13].

Multiple LBT requirement can be reduced by the concept of shared channel occupancy time (COT). In LAA, the maximum COT (MCOT) defines the maximum time allowed to share the channel among an end and its served nodes. This means a node/eNB that performs an LBT can occupy and share the channel for a specific period with served nodes/eNB. Nonetheless, a gap of 16 μs [44] is allocated between the transmissions of different nodes. If a gap is sensed over this limit, the node is further allowed to occupy the additional channel time without channel sensing. Most importantly, the rules require that the total channel occupancy by the initiating node and serving eNB, i.e., DL and UL transmissions, shall not exceed the corresponding MCOT. Thus, the concept of shared MCOT between the eNB and its UEs reduces the occurrence of situations where two exponential back-offs are needed before a UL transmission can happen. In this case, an eNB can initiate the transmit opportunity with the grant transmission (without LBT), and UEs can benefit from using reduced LBT as long as their transmission falls within the same MCOT, as shown in Figure 3b.

For example, in Figure 3b, an MCOT length of 4 ms is used. At the starting point, eNB begins DL transmission after an exponential random back-off is successfully achieved. The eNB sets a 3 ms DL transfer and leaves 1 ms for the UL before the MCOT reaches its maximum period. As there is a time gap between the last DL transmission and the first UL subframe, the UE only conducts a 25 μs CCA(clear channel assessment) LBT because it is still part of the eNB’s initiated transmit opportunity. As the previously distributed opportunity (i.e., 3 DL subframes + 1 UL subframe) exceeded the MCOT limit, a new LBT should be followed for a new transmission. In this regard, the proper allocation of the number of UL and DL frames in MCOT is still a challenge for a stable network.

### 4.2. Challenge 2: Seamless Connection due to Small Cell Architecture

When operating an NR-U network with mobility in the unlicensed bands, several potential problems arise from the combination of low transmit power requirements, coexistence requirements, and device mobility. The low transmit power of nodes will cause the cell sizes to become relatively small. In contrast, device mobility might cause the system to have a short time to handle the entire handover procedure when UEs move towards a cell having better link conditions. On top of this, the LBT procedure that is required for coexistence may cause a blocking of the transmission from either the eNB or the UE, which results in lost or delayed messages and delayed/outdated measurement reports. Having delays during the handover procedure in such small cells can potentially cause the UE to be out of coverage of its original source cell before it can complete the handover towards a target cell. Furthermore, this condition becomes more severe when mobility from a standalone LTE in the unlicensed spectrum to a standard LTE network on the licensed spectrum is expected to be seamless.

The current handover procedure is controlled by centralized eNB in LTE [45,46]. However, as we are operating solely in the unlicensed band, seeing the aforementioned challenges of small cell deployment and its regulatory requirements, a centralized eNB controlled handover procedure is not feasible. Thus, we need some advanced user-controlled handover similar to WiFi [47], where users are pre-configured with one or more potential target cells. Upon certain conditions being met, the UE may autonomously contact the target cell without informing the current cell. The pre-configuration of the UE for autonomous handover may be based on reported measurements of nearby cells such as the signal-to-noise ratio (SNR). In addition to this, we also can implement handover decisions to skip frequent transfers for fast-moving nodes such as UAV vehicles, warehouse robots, etc. [48,49]. This skipping decision can be achieved based on the upcoming cell’s topology; we can consider three criteria: (a) the area of the cell, (b) the trajectory distance within the cell, and (c) the distance of the BS from the cell edge. When a cell area is small, the sojourn time of the user inside that cell is limited; hence, the time for two successive handovers is short. Similar to the cell area, trajectory distance within the cell attempts to prevent frequent handovers. This option is more significant because it covers the situations where the area is wide, but the distance inside the cell of the consumer is small. The final criterion is the perpendicular distance between the BS of the upcoming cell and the boundary, which will be crossed by the user. This distance determines how close the current and next cell’s BSs are. Therefore, connecting to either BS will carry identical path loss results. Thus, as shown in Figure 4, the cell having less trajectory (red dotted line) can be skipped based on the combination of the above decision parameters. Here, we can use the TOPSIS [50] method, which is a multi-criteria decision analysis method to choose the correct option, integrated with the Q-learning [51] approach to provide intelligence in the system for managing a good QoS during the skipping phase (to find the right time). However, retaining the upcoming cell’s topology information for the user is still challenging.

### 4.3. Challenge 3: Neutral Host Deployment

The neutral host network (NHN) [52,53] access mode is one of the new network architectures envisioned by 5G. The NHN mode is a new self-contained network, which enables access authentication with or without a SIM card to provide services for subscribers from different types of service providers, including traditional mobile network operators, as well as non-traditional participating service providers. In a smart factory where the network is mostly for the robots and UAVs, but the factory also needs to support employees and visitors’ smartphones from different host providers, if we see the channel access mechanism proposed with Category 4 for NR-U, there are only four priority access labels available for NR-U devices, as shown in Table 6. All of the accessible equipment in the smart factory will connect via the access Category 4 specification for the channel access mechanism. The problem with NR-U while interconnecting these devices is that when the mobile users use high priority applications such as voice or video applications based on Category 4, the other mission-critical applications manufacturing devices such as robots or UAV devices that are assigned to do time-critical tasks will be highly interrupted. That is a smartphone using high multimedia (non-official work) data that will make channels busy for mission-critical applications. The interruption in this time-critical function can incur disasters in the smart factory. Hence, we need to modify the existing Category 4 priority access mechanism and design a new category specification that reconsiders specifications and manages resource allocation based on mission-critical applications or tasks.

A self-organized network (SON) [54] encompasses solutions to self-configure and self-optimize a network. It was introduced in LTE to facilitate the deployment of a system and to allow for further performance optimization. Mobility robustness optimization (MRO) and mobility load balancing (MLB) are two popular SON features in LTE. Other SONs such as energy saving (ES), inter-cell interference coordination (i.e., ICIC), enhanced interference mitigation and traffic adaptation (eIMTA), and coordinated multipoint (CoMP) operation are also available in the LTE network. Considering the role that SON has played in LTE in helping operators deploy and increase the robustness of the LTE networks, these similar features must be included in standalone NR-U. MulteFire White Paper Version 1.01 [34] specifies the SON element for the NR-U access point. Hence, SON features can be implemented to classify the priority access between the devices and optimize the performance of the mission-critical application. For this purpose, a network slicing approach can be adopted, where network resources are packaged and assigned in an isolated manner to a set of user devices according to their priority categories.The width of network slices is dynamically optimized based on the available load in each slice, as well as adjustments of COT, CW, UL, and DL can be implemented in each slice for intra-slice priority optimization. SON features accelerated with machine learning techniques [33] can play a significant role in processing data to enhance the performance of SONs.

### 4.4. Challenge 4: Adaptive Back-off Mechanism for LBT

The FCC mandates NR-U to use the Category 4 channel access mechanism, which is considered to be less efficient than the LTE-U-based carrier sense adaptive transmission (CSAT) mechanism [55] in terms of radio resource utilization. NR-U uses an LBT in medium access control (MAC) protocols, which is a contention-based channel access mechanism. Many times, resources are wasted while contending for channel access [56]. Additionally, a blind increase and decrease of the contention window in LBT bound the performance of NR-U equipment to a limited number of contenders, thus affecting the end-user quality of experience in the presence of the dense user.

The LBT scheme is the typical and traditional carrier sense multiple access/collision avoidance (CSMA/CA)-like mechanism, as shown in Figure 5, which was introduced in IEEE 802.11 Distributed Coordination Function (DCF) [57] for fair coexistence in the shared channel. A randomly generated back-off value for the contention procedure was used in LBT. At the first transmission attempt, the user generates a uniform random back-off value N, from the contention window interval [0, CW], where CW is initially set to the minimum value CWmin. After each unsuccessful transmission, CW is doubled until it reaches the maximum value CWmax = $2^{m}$ CWmin for *m* maximum number of back-off stages (M), that is *m*∈ (0, M). Once there is a successful transmission of its data frame, CW is reset to the minimum value CWmin. For a network with a heavy load, resetting *m* to zero and the contention window (CW) to its CWmin value after successful transmission will result in more collisions and poor network performance due to an increase in probability to select a similar back-off value *m* by the devices [58]. Similarly, for fewer contending UEs, the blind exponential increase of CW for collision avoidance causes an unnecessarily long delay due to the broader range for selecting CW. Besides, this blind increase/decrease of the back-off window is more inefficient in the highly-dense networks proposed for NR-U for 5G smart factories (with one million devices per square meter), because the probability of contention collision increases with the the increasing number of users. Thus, the current LBT protocol does not allow NR-U devices to achieve high efficiency in highly dense environments. Hence, to withstand this challenge, NR-U needs a more efficient and self-scrutinized back-off mechanism to promise enhanced user quality of experience (QoE) in Industry 4.0 applications.

The performance of an NR-U system can be severely degraded with an increase in the number of contenders, as the collision in the network is directly proportional to the density of the devices. Thus, we need a channel observation-based scaled back-off mechanism to scale-up and scale-down adaptively the size of the contention window to avoid the collisions [59,60]. The channel state information such as the estimation of channel collision probability or a number of users currently active in the shared channel is the best information for optimal selection of the CW value to utilize the resources more tightly. For the channel state estimation, detecting the time slot busy/ideal is a research challenge. Usually, the user senses the energy and compares its certain threshold to decide if the slot is busy or idle. However, it is very hard to distinguish only based on received energy. Thus, casting this question into a binary classification problem (K-means), unsupervised learning algorithms help to identify the state of a shared channel correctly and optimize the CW value.

### 4.5. Challenge 5: Multi-Domain Coexistence in a Shared Band

The two key bands that have emerged for NR-U are the 3.5 GHz shared access band (only in the U.S.) and the global unlicensed 5 GHz band. Military radar is the incumbent users of 3.5 GHz, whereas 5 GHz is popular among WiFi networks. In future IoT-based industries, there will be a wide range of devices and diverse applications based on WiFi, Bluetooth, ZigBee, etc., who will be sharing the same band with NR-U. Thus, NR-U must fairly share the available frequency resource with coexisting devices. The FCC proposed dynamic frequency selection (DFS) to protect incumbent users in the frequency domain, LBT, to share the unlicensed band in the time domain fairly [6]. However, these proposed protocols are limited to single domain applications. To the best of our knowledge, only a handful of research has worked simultaneously on multiple domain coexistence such as time, frequency, and space.

In this context, multiple domains can also be exploited for maximizing the fairness and spectral efficiency of the unlicensed band [8], as shown in Figure 6. We argue that for successful coexistence, the networks should possess flexibility in at least one of the above-listed areas. For example, WiFi is coexistence friendly as it exhibits agility in the time-domain, owing to its LBT operation. On the contrary, LTE (coexistence unfriendly) lacks this merit due to its fixed-length frames, and the nearest slot boundary is the earliest time it can react to coexistence-related challenges. However, one can introduce flexibility in other dimensions to compensate for the inflexibility in one dimension. A network with time-domain inflexibility such as LTE/LAA can add coexistence compensation in the frequency or space domain or both. In the above case, LTE can reduce coverage areas to combat the challenge of unfairness in a given area. In [51], the adaptive duty cycling (ADC) and dynamic channel switch (DCS) mechanisms were adopted. The joint time and frequency domain approach was proposed to support a fair channel access opportunity through muting (ADC mechanism) and increased efficiency of the spectrum through avoiding the most congested channel (dynamic channel switch). Moreover, the algorithm is reinforced by Q-learning techniques for support in a highly dynamic environment. A similar approach can be adopted for NR-U in shared channels. A more ambitious option is to exploit all available dimensions and increase the coexistence capability of a network. Note that a desirable property for a coexistence scheme is that it does not require substantial changes to the existing protocols and especially in the end-user hardware. Hence, it becomes more challenging to exploit integrated domains by using already available functions and components of technology.

### 4.6. Challenge 6: Coexistence of Asynchronous Access and Synchronous Access Systems

The unlicensed and shared band provides a simple, time-based spectrum access rule that is technology-neutral, fair to all spectrum users, and does not favor any unlicensed technology. However, when we consider the coexistence of asynchronous access systems with synchronized access systems for NR-U in the shared spectrum, synchronous systems can starve, as asynchronous systems could access the medium at any point in time and prevent equal access by synchronized access systems.

This problem can be eliminated entirely by using a dedicated spectrum band for the synchronous and asynchronous access systems, but for global adoption, this is quite infeasible. Hence, one of the possible approaches that can be used is conditional relaxation of the maximum channel occupancy time (COT). The conditional relaxation specifies the maximum COT for the asynchronous and synchronous system. In this approach, the maximum COT for asynchronous access systems is kept fixed while the COT for synchronous access systems is variable, i.e., the COT can be extended beyond the nominal value (only if the medium is utilized by synchronizing medium access during sync time). As shown in Figure 7, the extension of COT can be made shorter or longer to increase the efficiency of resources and to maintain fairness among the systems. On top of this mechanism, we also can use the multi-domain coexistence approach stated in Challenge 6 and the SON approach stated in Challenge 3 for COT optimization purposes.

## 5. Machine Learning as a Toolbox for NR-U

The realization of flexibility in any of the above challenges is not practical until the domains adapt to the dynamics of the coexisting environment, such as adjusting UL/DL, neighboring cell topology information, information of coexisting cells’ time, frequency domain, etc. The observations of the wireless medium and their performance or coordination with the neighboring system comprise one of the relevant research topics of optimization technology [8,61]. We can categorize coexistence scenarios into two: uncoordinated and coordinated schemes. In the former, networks implement coexistence mechanisms on their own without any consultation with their neighbors, whereas in the latter, systems directly or indirectly coordinate to ease their coexistence. The majority of the existing solutions are uncoordinated (local observation), as this does not require any infrastructure or change in the current network protocols. On the other hand, the coordinated solutions promise higher performance at the expense of higher complexity.

The uncoordinated systems look simple at first glance. Still, they require more sophisticated techniques to implement a neighbor-aware coexistence scheme due to the heterogeneity of technologies or the diversity in the ownership of the IoT devices in NR-U networks. Machine learning (ML) can provide a cognition toolbox to address the related problems. ML can enable identification (classification) schemes for recognizing coexistent communication systems and their basic rules of operation. For the “action” aspect, it yields adaptation-oriented intelligence, e.g., activation of transmission parameters such as time or frequency for NR-U coexistence or selection of spectrum mobility actions for specific coexistence contexts. ML classification allows the radios to identify the type of neighboring networks (e.g., an LTE or a WiFi network), channels, and their characteristics (addressing heterogeneity and lack of cross-technology communication). At the same time, reinforcement learning provides action-reward-based decision support in unstructured environments. Figure 8 below shows various ML techniques that can be used for NR-U systems to provide intelligence in the NR-U system. Table 7 gives a brief overview of the ML techniques and tools that can be used for the above-mentioned NR-U challenges.

Further, these proposed techniques also have some fundamental challenges for incorporating ML in 5G networks. The first one is complexity: low complexity is necessary for cost-efficient and practical systems. Secondly, the convergence time of decisions should be timely to provide tangible benefits for system operation. Additionally, the behavior of the coexistence scheme until convergence is vital since that may correspond to a significant portion of system time. Mobile or dynamic environments with transient responses causing a never converging ML brings the third learning challenge.

## 6. Conclusions

Licensed cellular networks are scarce and limited in flexibility, which is crucial for realizing the Industrial 4.0 5G vision. In this article, we argued that the unlicensed spectrum plays a key role in accomplishing the goals of the 5G Industrial communication goal. We investigated the NR-U technology for achieving industrial-grade 5G communication goals. Initially, we overviewed the Industry 4.0 revolution along with its essential performance requirements for communication networks. Further, we introduced NR-U as one of the key technologies in fulfilling industrial communication goals such as a wide bandwidth to support massive and diverse devices, ultra-high-speed for automation, low latency for mission-critical applications, ease of deployment, and improved reliability and security of wireless communications. We presented detailed challenges of NR-U due to regulatory requirements to operate in a shared/unlicensed band and the possible research directions. Lastly, we envisioned machine learning as a toolbox, which will play a fundamental role in making NR-U networks adapt to the coexistence environment.

## Figures and Tables

**Figure 1 sensors-20-02774-f001:**
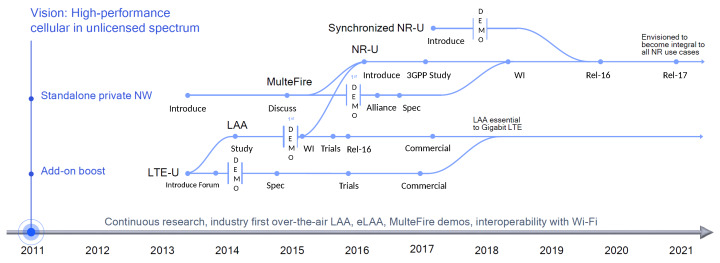
Evolution of NR-U [16]. eLAA, extended Licensed Assisted Access.

**Figure 2 sensors-20-02774-f002:**
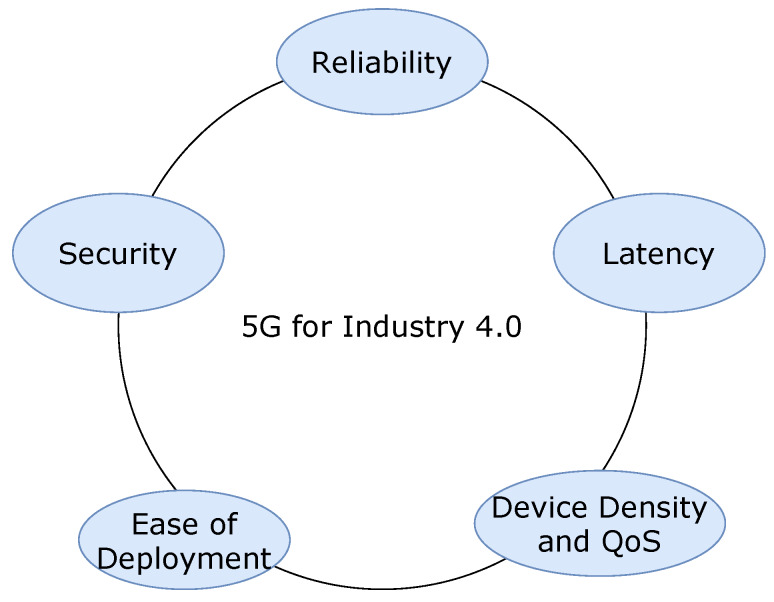
5G communication opportunity for Industry 4.0.

**Figure 3 sensors-20-02774-f003:**
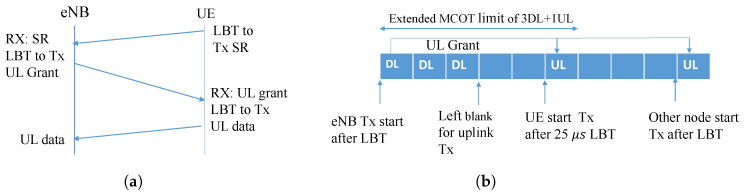
NR-U listen before talk (LBT) and shared maximum channel occupancy time (MCOT). SR, scheduling request. (**a**) NR-U LBT channel access procedure; (**b**) Shared MCOT.

**Figure 4 sensors-20-02774-f004:**
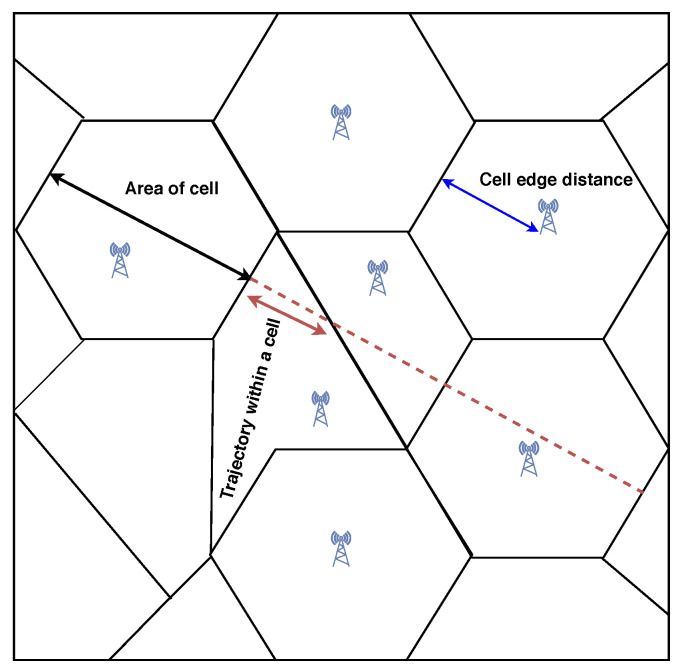
Handover skipping.

**Figure 5 sensors-20-02774-f005:**
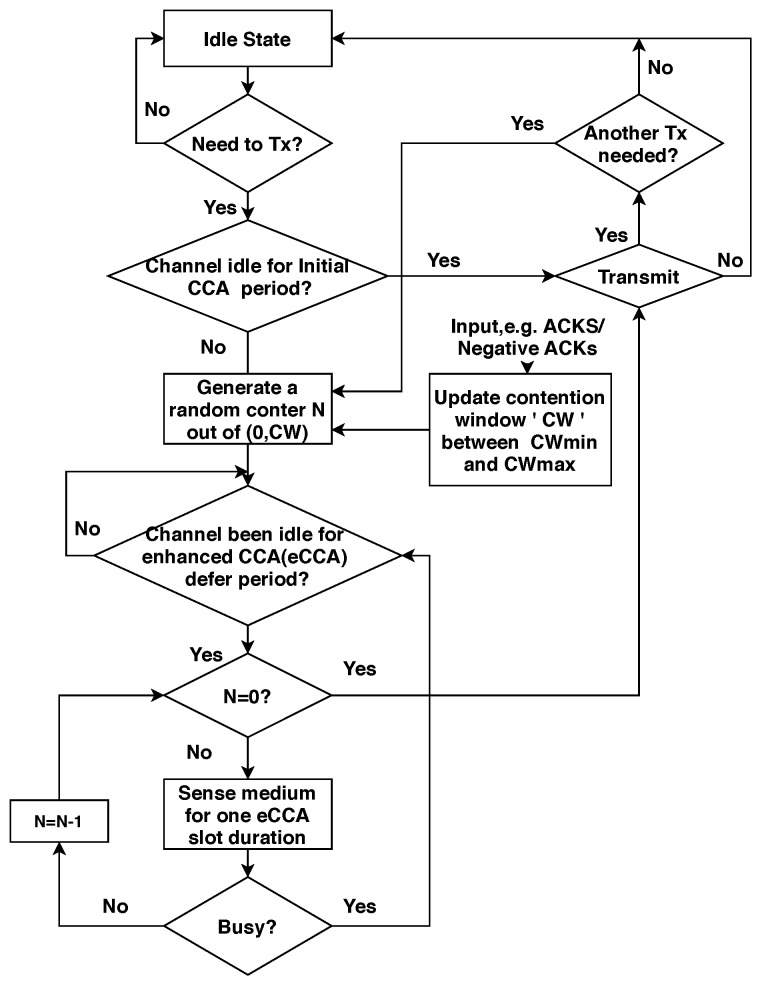
A flowchart of the LBT mechanism proposed by 3GPP.

**Figure 6 sensors-20-02774-f006:**
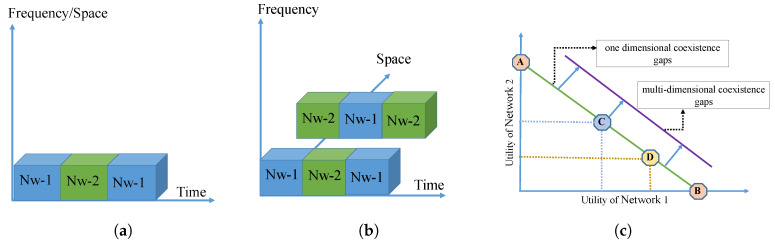
Multi-dimensional coexistence. (**a**) Coexistence gaps in one dimension (time domain); (**b**) Coexistence gaps in multiple dimensions (time and space domain); (**c**) The utilities of two coexisting networks.

**Figure 7 sensors-20-02774-f007:**
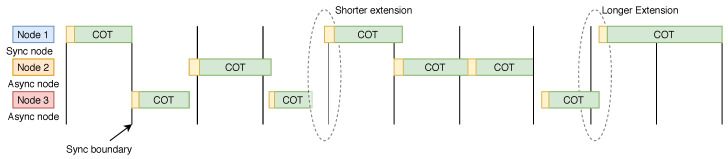
Coexistence of synchronous and asynchronous NR-U.

**Figure 8 sensors-20-02774-f008:**
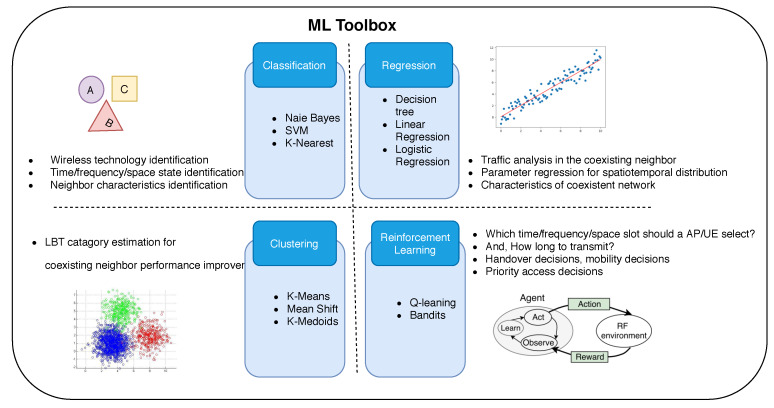
ML as a toolbox for NR-U.

**Table 1 sensors-20-02774-t001:** Use case of 5G NR for Industry 4.0 [14].

Use Case Type	Description
**Automation**	- Factory: floor robotics, e.g., wireless robots introduce significantly greater flexibility to reconfigure production lines. - Logistics and warehousing: e.g., pick and pack machines - Typically focused very dense deployments with low latency requirements
**Mission-Critical Services**	- To monitor and control critical infrastructure, e.g., electricity distribution grids, power plants, etc. - Public safety agencies often need to create closed user group ad-hoc networks at the scene of an emergency
**Primary Industries**	- Remote Industrial location often not covered by public wireless infrastructure - Very diverse sector for mining to agriculture, making increasing use of automated machinery

**Table 2 sensors-20-02774-t002:** Key performance requirement (KPI) of Industry 4.0 devices [27].

Industrial Devices	Latency	Availability	Throughput	Number
**Industrial robot**	<1 ms	>99.9999%	Kbps	>100
**Mobile robot**	<1 ms	>99.9999%	Mbps	>100
**Sensors**	∼100 ms	>99.99%	Kbps	>200
**Head-mounted display**	<10 ms	>99.9999%	G-Mbps	>50
**Handheld terminals**	<10 ms	>99.9999%	M-Kbps	>50
**Automated guided vehicles**	<10 ms	>99.9999%	Mbps	>10
**Security camera**	∼100 ms	>99.99%	G-Mbps	>10

**Table 3 sensors-20-02774-t003:** A taxonomy of the deployment modes of NR-U.

	Carrier Aggregation Mode	Dual Connectivity	Standalone Mode
**Standardization**	LTE Release 13	LTE Release 14	LTE Release 16
**Frequency**	5 GHz	5 GHz	2.4, 3.5, 5, 6, 37, 60 GHz
**Frequency band**	License + unlicensed	License + unlicensed	Unlicensed
**Protocol**	LTE	LTE	NR
**Aggregated bandwidth**	80MHz	80 MHz	800 MHz
**Streams**	Downlink	Uplink + Downlink	Uplink + Downlink
**Underlying technology**	LAA	eLAA	MulteFire

**Table 4 sensors-20-02774-t004:** Comparison of the asynchronous and synchronous modes of spectrum sharing. URLLC, ultra-reliable low-latency communications.

Mode	Asynchronous	Synchronous
**Name**	Evolutionary path of 5G	Revolutionary path of 5G
**Spectrum under consideration**	5 GHz	6 GHz
**Technology adoption**	Global	Under consideration (U.S. and Europe)
**Spectrum efficient**	No	Yes
**Spatial and** **predictable spectrum sharing**	No	Yes
**Synchronization of contention windows**	No	Yes
**Predictable latency**	No	Yes
**URLLC**	Not supported	Supported
**Time synchronization**	Not Needed	Needed
**Channel occupancy time (“COT”)**	COT ≤ 1msec	COT ≤ 6 ms COT ≤ 12 ms
**Coordinated multi-point (CoMP) mode gain**	Difficult to realize	High probability

**Table 5 sensors-20-02774-t005:** NR-U unlicensed/shared frequency.

Frequency	Adoption	Total Bandwidth	Incumbent Technology
**2.4 GHz [6]**	Global	100 MHz	Bluetooth, ZigBee, WiFi
**3.5 GHz [33]**	U.S.	150 MHz	Satellite, military radar
**5 GHz [37]**	Global	600 MHz	WiFi, WiGig, radar
**6 GHz [38]**	Europe	1200 MHz	Broadcast, fixed P2P and satellite service, cable TV relays
**60 GHz [10]**	Global	7 GHz	P2P fixed wireless bridging/backhaul

**Table 6 sensors-20-02774-t006:** Channel access priority of LBT.

Channel Access Priority Class (P)	Mp	CWmin,p	CWmax,p	Tm cat,p	Allowed CWp Sizes
1	2	3	7	2 ms	{3,7}
2	2	7	15	4 ms	{7,15}
3	3	15	1023	6 or 10 ms	{15,31,63,127,255,511,1023}
4	7	15	1023	6 or 10 ms	{15,31,63,127,255,511,1023}

**Table 7 sensors-20-02774-t007:** Summary of the challenges and the ML techniques. SON, self-organized network; CW, contention window.

Challenges	Problems	Methods	ML Functions	ML Techniques
LBT is mandatory to operate in the unlicensed or shared band	Latency due to LBT	Grant free transmission, shared MCOT	Identification and adaptation	Classification reinforcement
Seamless connection due to small cell architecture	Frequent handovers due to dense small cell deployment	Handover skipping	Detection and adaptation	Regression reinforcement
Neutral host deployment	Prioritizing mission-critical applications	SON/network slicing	Classification	Clustering classification
Adaptive back-off mechanism for LBT	Waste of resources and increase in collision probability due to contention-based access	Load-based adaptive CW approach	Detection and adaptation	Regression reinforcement
Multi-domain coexistence in a shared band	Most available coexistence solutions are limited to a single domain	Multi-domain solution approach	Adaptation	Regression reinforcement
Coexistence of asynchronous access and synchronous access system	Unfairness in resource	Relaxation of the maximum channel occupancy time	Identification and adaptation	Classification reinforcement

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
