# Peer review of "Future Is Unlicensed: Private 5G Unlicensed Network for Connecting Industries of Future"

_sensors, 2020, doi:10.3390/s20102774_

Round 1

Reviewer 1 Report

Dear authors,

this topic is very interesting, and I consider it for really original manuscript.

The paper has scientific soundness, but my methodical recommendations are:

Firstly, rework the abstract to persuade the readers about the reasons of significance of solved issue.

Secondly, to extend your Introduction and Literature review part I suggest:

Durana, P., Kral, P., Stehel, V., Lazaroiu, G., & Sroka, W. (2019). Quality Culture of Manufacturing Enterprises: A possible way to adaptation to Industry 4.0. Social Sciences, 8(4), 124. https://doi.org/10.3390/socsci8040124

Tuffnell, Caryl, Pavol Kral, Pavol Durana, and Tomas Krulicky (2019). “Industry 4.0-based Manufacturing Systems: Smart Production, Sustainable Supply Chain Networks, and Real-Time Process Monitoring,” Journal of Self-Governance and Management Economics 7(2): 7–12. doi:10.22381/JSME7220191

I hope my comment will be useful for your future work.

Reviewer 2 Report

The paper needs correction of some typos (e.g., page 4, undying; page 11, 3.5 MHz).

The authors fail to show what is the novelty of the paper. If the paper aims at being a tutorial it is lacking references, lacking a better description of the methods involved in answering the challenges and the authors ideas to address the problem.

If it is an attempt to propose a possible path to solve the problem of using NR-U for IIoT it lacks details on the methodology proposed for solving the challenges totally or partially as well as details on the ML and reinforcement learning choices to address the problem.

Author Response

Thank you very much for the detailed comments. Please find the attached.

Reviewer 3 Report

This paper is a survey about the future of private 5G unlicensed networks for connecting industries. The authors first give the introduction about Industry 4.0 and New radio in the unlicensed band(NR-U), then discuss carefully about the opportunities of NR-U for Industry 4.0, and then give six challenges in this research field. In the end they discuss the machine learning technique as a toolbox for NR-U. The paper is well organized, and truly give some interesting thinking about NR-U.
However, as a survey paper, I feel the references are insufficient. And the paper has some other problems. In Sec.2.1, they say "The advent of modern emerging industrial technologies, known as Industry 4.0, or Industrial Internet-of-Things(IIOT)". I feel the Industry 4.0 and the IIOT are two different concepts. though they may have some similar places. In figure 1, the words in this figure are too small to be seen clearly. Besides, this paper is good as a whole and could be published after a careful revision.

Author Response

(The authors gave the same response as above.)

Round 2

Reviewer 2 Report

The authors have been able to improve the quality of the manuscript and to address adequately the remarks made in the previous review.

Some English proofreading is needed if the manuscript is accepted. There are still typos to correct including in the new text marked in blue.